# Characterization of Natural and Synthetic Fabrics for the Treatment of Complex Wastes

**DOI:** 10.3390/polym16010084

**Published:** 2023-12-27

**Authors:** Alexis López-Borrell, Jaime Lora-García, Vicent Fombuena, Salvador C. Cardona, María-Fernanda López-Pérez

**Affiliations:** 1Instituto de Seguridad Industrial, Radiofísica y Medioambiental (ISIRYM), Universitat Politècnica de València (UPV), Plaza Ferrándiz y Carbonell, s/n, 03801 Alcoy, Spain; jlora@iqn.upv.es (J.L.-G.); scardona@iqn.upv.es (S.C.C.); malope1@iqn.upv.es (M.-F.L.-P.); 2Technological Institute of Materials (ITM), Universitat Politècnica de València (UPV), Plaza Ferrándiz y Carbonell 1, 03801 Alcoy, Spain; vifombor@upv.es

**Keywords:** natural fiber, brackish water, liquid absorption capacity, mechanical/thermal properties, degradation, capillarity, contact angle

## Abstract

In the present study, nine fabrics have been tested for brackish water treatment with the aim of industrial application under the concept of zero liquid discharge (ZLD). Moisture content was determined, where it was observed that the lignocellulosic fabrics had a moisture content ranging from 2.5 to 8.5%. The wetting contact angle showed that the flax with polylactic acid (LPLA) was the most hydrophobic. The determination of the liquid absorption capacity showed that, of the synthetic fabrics, the one with the highest absorption, both in distilled water and in brackish water, was the polyester (PES) fabric with an absorption of 816% compared to its initial weight. In the natural fabrics, the highest absorption capacity was shown by the wet-laid without treatment (WL-WT) fabric for both distilled water and brackish water, although it required several cycles of operation to maintain this stable absorption. Exposure to brackish water improved the absorption capacity of all samples. Mechanical and thermal characterization showed that the synthetic fabrics were more resistant than the natural fabrics, although they may compete in terms of applicability. The capillarity study showed that the most hydrophilic fabrics completed the test the fastest. Finally, the composting degradation test showed that those fabrics with polylactic acid (PLA) content degraded faster in the first 14 days and thereafter the degradation of the lignocellulosic content showed a slower degradation until 112 days. The Bam fabric did not degrade during the course of the experiment.

## 1. Introduction

Nowadays, with the growth of the world’s population, the consumption of resources is increasingly higher and requires a larger industry to satisfy the population’s requirements. Among the most affected sectors is the demand for fresh water and food. In recent years, freshwater production has been remedied with the increase in desalination plants, while the large amount of waste produced goes mostly to urban solid waste treatment plants [1].

In 2015, brackish and seawater desalination plants had a 97.5·10^6^ m^3^·d^−1^ world production capacity. There are currently more than 20,000 desalination plants in 150 countries and it is expected that by 2050 the production capacity of desalination plants will rise to 192·10^6^ m^3^·d^−1^. There are multiple freshwater production systems, such as multi-effect evaporation and membrane separation technology using reverse osmosis systems. It is known that desalination plants cannot achieve production yields of 100%, which is why this freshwater production generated a waste estimated at a volume of 141 Mm^3^ of concentrated brine in 2020 [1,2,3,4].

Another existing problem would be the concentrates collected from landfill leachate. These wastes have general characteristics such as high organic load, suspended solids (SS), nitrogen in the form of ammonium, and heavy metals. These components are affected by several factors, such as the site’s climate, the type of waste, or the age of the leachate [5,6,7]. Some conventional treatments applied to these wastes include biological processes, such as aerobic–anaerobic ones, and membrane technology. On the other hand, it is possible to apply chemical treatments such as coagulation-flocculation or mixtures of physical-chemical treatments for higher efficiency [8,9,10,11].

Zero Liquid Discharge (ZLD) is a proposed management method for treating concentrated leachate and brine wastes. This treatment method does not generate a water flow, but the waste remains in solid form and is much easier to manage. In addition to the simplicity of its management, in some cases, the waste can be of added value and can be used to generate a circular economy structure [12,13,14].

In order to work on the ZLD line, instruments are required that are affordable and as environmentally friendly as possible. In recent years, natural fibers have gained importance in this subject. The production of natural fibers worldwide is more than 25 Mt. For example, bamboo has an annual production of 10 Mt, flax of 0.81 Mt, and jute of 2.5 Mt. In addition, banana, abaca, pineapple, and wood fibers are also produced [15,16,17]. They are essentially composed of cellulose, hemicellulose, and lignin and some of them have a low percentage of waxes [18,19]. In some applications where the limitations of natural fibers are palpable, it is necessary to use synthetic fibers, including glass fiber, polyester, or carbon fiber, all of them with higher thermal and mechanical properties. The major difference between synthetic and natural fibers lies in their mechanical and thermal properties, with synthetic fibers generally being better.

On the other hand, to use natural wastes, such as wood waste, non-woven fabrics are manufactured using the wet-laid technique. The manufacturing process of fabrics involved in the wet-laid technique is similar to the production process of the paper industry, but in the wet-laid case, the lignocellulosic residue is mixed with the highest possible percentage of wood pulp and a compatibilizer that can hold the fibers together and give mechanical properties to the material [20,21]. Fibers with a high cellulose content have a very low contact angle with pure water, which allows their surface to have a high affinity to water and, therefore, show good water absorption. On the other hand, surfaces with a high contact angle to pure water are considered non-wettable, such as polyethylene or polyester. Thus, both types of surfaces could be defined as hydrophilic and hydrophobic samples, respectively [21,22].

Therefore, due to the increased environmental awareness and the application of new processing techniques, such as the wet-laid technique, publications related to natural fibers are extensive [23,24]. However, its use for the treatment of water and brackish is still very scarce.

This study’s main purpose is to characterize five fabrics of natural origin and four synthetic fabrics to be applied in brackish water treatment. The moisture content of each fabric and the water contact angle will be characterized to determine its hydrophilic-hydrophobic character. In addition, key aspects such as the capacity for water, brackish water absorption in different cycles, and the capillarity of the fabrics will be studied. Technical properties that will determine its scalability at an industrial level will be studied through mechanical tests, tensile tests, and thermal properties through thermogravimetry (TGA). Finally, a compostability study under thermophilic conditions will also determine the biodegradability of natural fabrics.

## 2. Materials and Methods

### 2.1. Fabrics Used in the Research

For the experimental development, nine fabrics have been used; five of these samples are from natural origin and the rest are synthetic fabrics. Natural fabrics include jute (Jut), bamboo (Bam), flax with a 50–50% mixture of polylactic acid (LPLA), and two non-woven fabric developed using wet-laid techniques (WL). These two wet-laid fabrics are composed of a mixture of 70% palm tree pruning, 10% lyocell, and 20% PLA. The difference between them is the application of a thermal treatment through calendaring for the fusion of the polymeric part and to provide greater cohesion between the components, in one case (WL-T), and without treatment in the other case (WL-WT). Both wet-laid fabrics were supplied by Aitex (Alcoy, Spain). The main characteristics of these five natural fabrics can be seen in Table 1.

On the other hand, the synthetic fabrics used in the experiments are commercially available fabrics consisting of non-woven (GF-NW) and woven glass fiber (GF-W), polyester (PES), and aramid (Ara) fabrics. These fabrics were supplied by Castro Composites (Pontevedra, Spain). The type of fabric, thickness, and grammage of each synthetic material used can be seen in Table 2.

### 2.2. Salts Used in the Research

For the experiments, salts of sodium nitrate (NaNO_3_) were supplied by Scharlau Chemie S.A. (Barcelona, Spain), while potassium nitrate (KNO_3_), disodium tetraborate (Na_2_B_4_O_7_·10H_2_O), calcium chloride (CaCl_2_), calcium hydroxide (Ca(OH)_2_), magnesium sulphate (MgSO_4_·7H_2_O), sodium hydrogen carbonate (NaHCO_3_), copper sulphate (CuSO_4_·5H_2_O), and lithium chloride (LiCl) were delivered by PanReac Applichem, ITW Reagents (Barcelona, Spain). All these inorganic solutions will be used at different concentrations to prepare a synthetic solution of brackish water. The concentrations of the salts used in the current research can be seen in Table 3.

### 2.3. Humidity Determination

The humidity of the different fabrics from natural and synthetic origin was determined by using the difference in mass before and after being dried. Fabric samples were cut with a surface area of 100 × 100 mm^2^ and their initial mass was recorded on an analytical balance, Nahita blue series 5134. A minimum of five tests have been conducted per fabric and the average humidity value and deviation are shown as the final result. Samples are placed in a drying oven at 105 °C for 2 h. After this period, samples are placed in a desiccator until they reach room temperature. Once they have reached this temperature, the sample is weighed on the precision balance and its mass is recorded. The fabrics are returned to the heater for 1 h. This process is repeated until the difference in weighing the samples consecutively is less than 0.1%.

The equation used for the calculation of the humidity content is shown below:(1)$$w_{H_{2}O\;}{}_{}\left(\%\right)=\frac{m_{0}-m_{1}}{m_{0}}\cdot 100$$
where $w_{H_{2}O\;}$ (%) is the humidity content of the sample, $m_{0}$ (g) is the mass of the sample before being subjected to drying cycles, and $m_{1}$ (g) is the mass of the sample at constant weight after drying.

### 2.4. Wettability Characterization

The interaction of the surface of the samples with the test liquid was measured by monitoring the wetting angles using an EasyDrop Standard goniometer, model FM140, supplied by KRÜSS GmbH (Hamburg, Germany). The equipment can measure contact angles from 1 to 180° with an accuracy of ±0.1°, using analysis software (Drop Shape Analysis SW21; DSA1).

In order to measure the contact angle in the equipment, the samples of the different fabrics were cut with a surface area of 25 × 25 mm^2^. A drop of distilled water is deposited on their surface using a microsyringe. The contact angle is measured continuously until the drop is completely absorbed into the sample. At least ten measurements by drop and five different drops of distilled water were carried out for each fabric, and average values were calculated. The maximum error did not exceed 3%. The contact angle measurement over time (∆*Θ*·min^−1^) was determined. To evaluate this value, the initial contact angle (*Θ*_0_), the final contact angle (*Θ_f_*), and the time elapsed in the test were measured. The experiment ended when the water droplet was completely absorbed or its contact angle decreased by at least 30°.

### 2.5. Liquid Absorption Capacity

To determine the liquid absorption capacity (*LAC*) of the mentioned fabrics for distilled water and for a solution of brackish water, the procedure described in the UNE-EN ISO 9073 [30] standard was followed. Briefly, a minimum of five specimens of each fabric of 100 × 100 mm^2^ were weighed to determine the mass of the dried sample, $m_{k}$ (g). The variable $m_{n}$ (g) is the combined mass of fabric and liquid absorbed after the test for each of the five samples. For this procedure, the sample was submerged in the liquid for a period of 60 ± 1 s. The liquid column above the fabric’s surface should be at least 20 mm during this time. Once this time has elapsed, the fabric is extracted from the problem liquid with the least possible manipulation and without draining it. It is held perpendicularly for 120 ± 3 s so that it loses the excess liquid that is not retained by the sample. After this time the sample is weighed to determine $m_{n}$ (g).

The following expression was used to determine the *LAC* and the average value was shown as the final result:(2)$$LAC=\frac{m_{n}-m_{k}}{m_{k}}\cdot 100.$$

The maximum liquid absorption capacity (*LAC_Max_*) was also determined by immersing the samples in distilled water for 48 h to compare their maximum liquid retention capacity, since with the described standard process the fabrics are immersed only for 60 ± 1 s and may differ with respect to the maximum capacity of the samples.

### 2.6. Liquid Absorption Capacity after Five Cycles

To observe the behavior of the fabrics during several absorption cycles, the above-mentioned experimental procedure was carried out with distilled water and with the brackish water solution with slight modifications. In order to be able to use the same fabric sample for five cycles, a drying process at 50 °C in an air oven for 24 h was carried out between consecutive tests. After this period, the samples were deposited in a desiccator until they reached room temperature, thereby making them available for the next cycle.

### 2.7. Morphological Characterization

The surface of the samples was observed using a field emission scanning electron microscope (FESEM), the ZEISS ULTRA model from Oxford Instruments (Abingdon, UK). Morphological characterization of the samples was carried out for the fabrics before and after exposure to the brackish water dissolution. Before the samples were observed in the FESEM, they were coated with a thin carbon layer by using a sputter-coater EM MED020 from Leica Microsystems (Wetzlar, Germany). Once the samples were prepared, they were operated at an accelerating voltage of 2 kV.

### 2.8. Mechanical and Thermal Characterization

Tensile tests were performed using a universal test machine Elib 30 (Ibertest S. A. E, Madrid, Spain), according to the guidelines of the ISO 9073. Five specimens per fabric with a length of 150 mm and width of 100 mm were tested using a crosshead speed of 300 mm·min^−1^ and a load cell of 5 kN. Average values of maximum strength (MPa), elongation at break (%), and tensile modulus (MPa) were calculated.

To determine if the tensile properties were modified when the fabrics were wet, samples were immersed in distilled water for 24 h, to ensure its complete wetting. After this time, the samples were removed from the water bath and subjected to tensile tests in the same way as previously described.

Thermogravimetric analysis (TGA) were carried out to determine the initial and final degradation temperature (*T_onset_* and *T_endset_*, respectively) and the lost weight at maximum degradation temperature (∆*wt*). The experiment was performed in two steps, the first to analyze the humidity content of the samples and the second to analyze their degradation. The process temperature selected for the analysis of the samples was from 30 to 700 °C in a nitrogen atmosphere with a flow rate of 66 mL·min^−1^ and a heating rate of 20 °C·min^−1^, on a TGA/STDA851 thermobalance from Mettler Toledo Inc. (Schwerzenbach, Switzerland).

### 2.9. Sample Capillarity

To determine the capillarity of the fabrics, samples of 50 × 150 mm^2^ were immersed 15 mm perpendicular in the bath containing the test liquid. To measure the rising height of the liquid, the zero height is taken at the free liquid surface and this is increased as the assay progresses. The values of the liquid rising through the fabric are taken every 5 min. The capillarity test through the fabrics ends when the liquid column has reached 135 mm. If it does not reach the maximum height of the samples, it is left in the bath for 48 h to determine the maximum height it can achieve.

### 2.10. Degradation under Composting Conditions

The composting disintegration test was performed, adapting the normative ISO 20200 [31] to polymeric samples. These tests were carried out under aerobic conditions in thermophilic conditions (58 °C) with a relative humidity of 55%. Seven different samples of each fabric (25 × 25 mm^2^) were previously dried in an air oven for 24 h at 40 °C to remove the humidity. The dried samples were buried in a synthetic compost, prepared according to the normative, and placed in a reactor. To control the process, different samples were unburied periodically on days 14, 30, 47, 62, 76, 93, and 112. After extraction, the samples were washed with distilled water to remove the composting matter adhering to them and dried in an oven at 50 °C for 24 h before weighing. The percentage of weight loss (*W_L_* (%)) of the extracted samples is calculated from the following equation:(3)$$W_{L}=\frac{w_{0}-w}{w_{0}}\cdot 100.$$

## 3. Results

### 3.1. Humidity in Fabrics

In this section, the humidity content of the natural and synthetic fabrics mentioned above was determined. The results are shown in Figure 1.

As shown in Figure 1, fabrics with higher lignocellulosic fiber proportion have higher humidity content. Jut fabric stands out with a humidity of 8.58%, followed by WL-T and WL-WT fabrics with humidity content of 5.17% and 4.28%, respectively. Meanwhile, Ara, Bam, and LPLA fabrics have similar values of around 3% humidity.

Studies on natural fibers have shown that the chemical composition of these compounds is mostly cellulose and hemicellulose. These components present OH radicals in their molecular structures, giving the possibility of a high degree of affinity to form hydrogen bonds with water and to present these high moisture contents. The moisture content obtained in the present study for lignocellulosic fabrics is similar to that demonstrated by other authors. Kumar et al. [32] compiled data on moisture content, reporting 10% for flax and jute fibers and 11.7% for bamboo. On the other hand, Gholampour and Ozbakkaloglu [16] reported a moisture content of 11% for sisal fibers, similar to WL-WT and WL-T samples. The moisture values of the present study are similar to those reported in the literature, although these values can be affected by environmental factors of temperature and ambient humidity, as mentioned by other authors [33]. As observed in this study, there may be a direct relationship between moisture content and its application in treating brackish water. Due to their nature, lignocellulosic fibers reach equilibrium with the surrounding environmental conditions. As a result, the ability to absorb liquids such as brackish water can be limited under high ambient humidity conditions. The GF-NW, GF-W, and PES synthetic fabrics have less than 0.1% moisture content. In the case of moisture determination for the PES sample, it is an organic polymer with no functional groups that would allow interaction with ambient humidity. The GF-NW and GF-W samples are based on SiO_2_ polymer chains with a stable, centered tetrahedral structure, significantly limiting the interaction with ambient humidity. The results obtained in the moisture determination for these samples are similar to those demonstrated by Valipour et al. [34].

### 3.2. Wetting Contact Angle Measurements

Table 4 summarizes the results of the initial angle (*Θ*_0_) when the drop of water is deposited on the fabric’s surface, the final angle (*Θ_f_*) during the time elapsed in the test, and the variation of this angle over time (∆*Θ*).

Ara and Bam samples can be highlighted due to their rapid absorption in the experiment. When the drop of distilled water was deposited on these two samples, it was immediately absorbed upon contact with the surface. In the case of the Bam fabric, its absorption was so fast due to its low specific weight, as the thickness of the sample is 0.2 mm. The Ara sample absorbs the water droplet very quickly, probably due to its chemical structure as it is composed of aromatic polyamide chains with polar functional groups.

In the GF-NW sample, the distilled water droplet deposited on the fabrics passed through the fibers due to the unidirectional weave type. Its fibers are arranged in a 90° weave and the weave that joins them is scarce, therefore the drop of distilled water passes through the fibers and is not static on them. The GF-W fabric showed that when the distilled water droplet was placed on its surface, the contact angle was less than 30°, indicating that the liquid droplet has been dispersed on its surface. Lim and Lee [35] studied the wettability of glass fibers treated from plasma in an oxygen atmosphere at different powers and at different exposure times. These fibers before plasma exposure showed a contact angle with distilled water slightly higher than 25°, thus determining that the water droplet’s interaction with the fabric’s surface causes it to disperse.

The Jut fabric shows a warp between each of its threads at a 90° angle, forming a weave with greater cohesion than GF-NW, and the drop of distilled water can remain on the surface of the weave. In this case, the initial angle has a value of 156.5°, indicating that its interaction with water is low, but the drop is wholly absorbed in a time of 4 min. Sever et al. [36] reported the contact angle for jute fibers with different treatments, showing that the untreated jute fibers had low surface energy, demonstrating their low interaction with water and high initial contact angle.

The WL-WT and WL-T samples are made from palm pruning (70%) with chemical compositions similar to natural fabrics but blended with 20% PLA for increasing mechanical properties. The presence of this polymer reduces the ability to absorb the distilled water droplet on the fabric’s surface. The difference between the two fabrics is due to the thermal treatment that one of them has received, showing a lower thickness and, therefore, its surface roughness decreases. As its surface roughness decreases, the water droplet has a larger interaction surface but its absorption rate is limited because there are no empty voids to facilitate the passage of water into its structure as in the case of the WL-WT sample.

The PES sample is a synthetic fabric made from polymeric chains with low interaction with water, as its structure is based on organic chains without polar functional groups. Ghenaim et al. [37] studied polyester fibers treated with a graphite surface to increase their interaction with water. In this research they determined, for untreated polyester fibers, a surface energy lower than that of water, which indicated a higher initial contact angle and non-interaction with water. The decrease in the contact angle is due to the evaporation of the liquid.

Finally, the LPLA sample is composed of 50% flax and 50% PLA, showing an initial angle of 152.7° with a decreasing contact angle behavior as PES. Erzsébet et al. [38] investigated the wettability of flax fibers treated with plasma to reduce their initial contact angle. Their research determined that the surface energy of flax fibers was lower than that of water and, therefore, their initial contact angle was higher. This phenomenon, added to the effect of PLA on their structure, caused a decrease in their interaction with water. According to this fact, we guess that the water droplet deposited on the sample’s surface is evaporating. Therefore, considering the high contact angle exhibited by the LPLA fabric and the extended residence time of the formed droplet, the authors deem this fabric highly hydrophobic. Consequently, the interaction of water with the LPLA fabric is limited, which may result in low absorption coefficients for brackish water.

Natural fabrics share in their chemical structure different proportions of cellulose, hemicellulose, and lignin essentially. Cellulose has -OH groups, which are mainly responsible for the formation of hydrogen bonds. Hemicellulose does not have a defined structure compared to cellulose, so hemicellulose can interact better with adjacent molecules, which are mainly responsible for its hydrophilic behavior, as mentioned [39,40]. Compared to cellulose and hemicellulose, lignin has aromatic groups in its structure, making it more hydrophobic than the other two components, as reported by Hollertz et al. [41]. In the case of synthetic fabrics, such as PES or natural fabrics with PLA blends, they are based on polymers that contribute to hydrophobic behavior unless their surface is treated in some way to increase their surface energy, as shown by Jordá-Vilaplana et al. [42].

As can be deduced from the results, the Ara and Bam samples would be the most interesting for an industrial application due to their high degree of interaction with water by completely absorbing the drop deposited on their surface, indicating the high degree of wettability in such a short contact time. On the other hand, the fabrics with the worst wetting behavior for an industrial application would be LPLA and PES as their variation in contact angle over time is very slow, indicating their low surface interaction with the liquid in contact.

### 3.3. Liquid Absorption Capacity in Distilled Water

The absorption capacity of each of the above-mentioned fabrics was determined. Figure 2 shows the results for each of them.

As can be seen in Figure 2, the PES fabric stands out above the rest with an absorption capacity of 816%. This fabric is often used as a home textile for drying surfaces due to its high absorbency. This may be due to the fact that water is embedded in its macroporous structure.

This water absorption into the fabric structure is due to a difference in concentration. When the fabric is forced to soak in the liquid for 1 min, the water concentration on the outside of the fabric is complete, while inside the fabric structure there is only a concentration of water associated with the equilibrium moisture content of the material. Once it has been immersed in the water, this gradient of concentrations causes the water on the outside of the fabric to displace the air contained between the holes in the fabric, producing water absorption.

The WL-WT and WL-T fabrics present variations mainly due to the heat treatment to which one of them has been subjected. In the case of no treatment, the structure presents a slightly higher thickness, 2 mm, than the one with heat treatment, 0.5 mm. This difference causes the distilled water to have a lower absorption in the fabric with less thickness (WL-T) than its other variant (WL-WT). Despite this, the WL-T fabric has better mechanical behavior, as the WL-WT had to be manipulated with great care because it broke with the little handling carried out. Fook and Yatim [43] presented water absorption values over time from 250 to 350% for untreated pineapple leaf fibers. In this study the WL-WT and WL-T samples are mainly from palm leaves giving similar values to those mentioned by Fook and Yatim.

Bam fabric has the lowest grammage compared to the rest of the fabrics, as its thickness is 0.2 mm. In this case, it presents a higher absorption behavior to distilled water than the rest of the fabrics, except for PES and WL-WT. In the tests carried out, it was observed that as the sample has such a small thickness it is completely wetted as soon as it came into contact with the liquid, presenting its maximum liquid absorption capacity at the end of the 1-min test.

The other fabrics with the best liquid absorption characteristics in distilled water are LPLA, Jut, and Ara. These samples present liquid absorption values of the same order of magnitude, although they differ slightly.

Finally, the two glass fiber fabrics do not absorb distilled water in the same way as the other fabrics. In these, the water remains embedded between the fiber bundle (GF-NW) or is retained on its surface (GF-W) as it is more compact. In this way, there is an increase in mass due only to the water that is retained on the structure and not inside it like the rest of the fabrics. Glass fiber is often used as a reinforcement material for composite materials and not for direct use as reported in this research for the GF-NW and GF-W samples. Araújo et al. [44] show that for composites with a glass fiber content of up to 40% water absorption is around 1% by immersing the composites for a long time in water.

In view of the results obtained, the GF-NW and GF-W samples are not interesting for industrial application due to their low water absorption. On the other hand, the PES fabric presents the highest absorption capacity in the contact time with a value of 816%, which is of greater interest. Finally, the Bam fabric is one of the most interesting, as its absorption capacity is very high compared to its thickness, which would facilitate the evaporation of water, making it of interest for its industrial application.

### 3.4. Liquid Absorption Capacity in Five Operation and Drying Cycles in Distilled Water

As mentioned above, the fabric samples have been tested through five wetting and oven-drying cycles. The aim is to observe if the behavior of the fabrics is modified throughout the working period, simulating an industrial process of cyclic operations.

The first treatment has been carried out with distilled water. The aim is to observe if the voids between the fibers are rearranged when the inclusion of water displaces them. Moreover, it is studied whether the water is absorbed more easily after drying as these hollows are already adapted to the passage of water and, therefore, may improve the absorption of liquid when it is in contact with the fibers. These results can be seen in Figure 3.

Figure 3 displays the five cycles performed on each of the fabrics and compares them with their *LAC_Max_*. This maximum capacity was obtained by immersing the samples in distilled water for 2 days.

For WL-WT, WL-T, LPLA, and Jut fabrics, the absorption capacity increases as the number of cycles increases, although it still differs from its maximum absorption capacity. This effect can be due to the empty spaces between the fibers of the fabrics, depicted in Figure 4, which shows the FESEM images of all the fabrics tested. In the case of wet-laid non-woven fabrics, a more significant number of empty voids can be observed, in which the fabric can occlude water and retain it in its structure.

Analyzing the FESEM images for WL-WT and WL-T, depositions can be seen on the WL-T fabric. These deposits are due to the heat treatment applied. Nevertheless, the WL-WT fabric shows smoother yarns and the palm fibers are larger and with a rougher structure. On the other hand, Jut and LPLA fabrics have greater cohesion between their fibers than non-woven fabrics. So, their *LAC* is lower, although water is able to displace these empty spaces and increase the absorption capacity as the work cycles increase.

The fabrics of Bam and Ara have shown no significant changes over the operating cycles. Compared to the rest of the fabrics, their fibers are tidy. In the case of the Bam fabric, its grammage is 56 g·m^−2^ with a thickness of 0.2 mm. Observing the FESEM images, it is possible to appreciate the hollows between its woven fibers, where the water is retained when the fabric is submerged for the liquid absorption tests. These small spaces do not allow the deformation of the voids to keep more water as the cycles increase. In the case of the Ara fabric, its grammage is higher than that of Bam. When observing the images in Figure 4, it can be seen that the empty spaces between its fibers are smaller, but its absorption capacity is similar to its *LAC_Max_* because this type of fabric has terminal groups in its chemical structure that are similar to water. These two fabrics would be of industrial interest due to their stability in water absorption over operating cycles.

The GF-NW and GF-W samples show similar behavior to Bam and Ara, as their *LAC* does not change over the operating cycles. As they are woven and non-woven glass fibers, they practically do not absorb water, but it is retained on their surface. Ramamoorthy et al. [45] conducted water absorption studies on single-fiber, reporting that glass fiber absorbed about 2.5% after ten days of immersion in water. Analyzing the FESEM images in Figure 4 for the GF-NW and GF-W samples, there is no gap between their fibers to retain water, so the water absorbed in the test is due to water being held on their surface and not between their fibers. These two samples would be ruled out for industrial use due to their low absorption capacity.

Finally, the case of the PES fabric is the one with the highest absorption capacity. Therefore, when submerged for only 1 min, it presents very high deviations, where no clear tendency can be seen with this type of fabric. By observing the images of the FESEM, one can appreciate its non-woven structure and the number of empty spaces between its fibers that retain the water embedded in its structure. This fact, together with its thickness, allows it to retain a high amount of water. This fabric could be interesting for industrial applications due to its high capacity to absorb liquids.

### 3.5. Liquid Absorption Capacity in Five Operating and Drying Cycles Using Brackish Water

The absorption and drying cycles when the problem liquid is a synthetic brackish water solution are shown in Figure 5, comparing the cycles with the *LAC_Max_* obtained with distilled water. Only liquid absorption is considered, discounting the salts retained in the fabric. To discount the weight of the salts retained in the fabrics for calculating *LAC* using Equation (2), in a similar way when only distilled water is used, the following procedure has been used. The weight of the fabric together with the precipitated salts has been obtained after drying the samples. Taking into account the weight of the fabric before being exposed to the brackish water solution, it is possible to determine the weight of salts for each particular work cycle using a simple subtraction. This value has been used for calculating $m_{k}\;\mathrm{and}\;m_{n.}$ The fabric used for each working cycle is not cleaned, so it keeps the salts retained in its structure, except for those that are dissolved or released during the next cycle. On the other hand, the brackish water solution is renewed between every five samples and between work cycles to keep the concentration of the solution as stable as possible.

As can be seen in Figure 5, the presence of salts improves, in all cases, the water absorption in the fabrics concerning the *LAC* values in Figure 3. In the case of the WL-WT sample it decreases slightly while the WL-T sample increases slightly over the cycles. This increase in *LAC* in the samples may be due to the deposition of salts in its porous structure, as shown in Figure 6. According to the FESEM images, salt deposition on the WL-T fabric is more severe than on the WL-WT sample, probably because it is a more compact structure and, therefore, the voids between fibers are smaller. It is expected that these salt deposits help to retain the water inside the fabrics.

The GF-NW and GF-W samples do not show a difference in behavior with respect to the results observed in Figure 3 with distilled water, probably due to its lower thickness and lower retention of salts. This tiny salt deposition can be seen in Figure 6, where in both cases it is produced superficially and does not penetrate into the compact fiber structure, as seen in the red box marked for the GF-W sample.

In view of an industrial application, WL-WT and WL-T fabrics would be of greater interest due to their stability in liquid absorption. Nevertheless, the GF-NW and GF-W samples show the same behavior as when exposed to distilled water, which would rule out their industrial use.

The Bam and Ara samples have shown to be fabrics with an absorption close to their maximum capacity throughout the experiments. This is observed when comparing the results with brackish and distilled water, where the *LAC* values have not significantly altered. There is a clear difference between these two fabrics, since the Bam sample has a thickness of 0.2 mm and the Ara fabric has a thickness of 1 mm. Comparing both fabrics in Figure 6, it can be seen that the deposition of salts in the Ara sample, marked in red circles, is more severe and occurs between its fibers compared to the Bam sample, where the deposition is superficial. Due to its low grammage there is little surface available to form salt crystals.

On the other hand, the Jut and LPLA fabrics show a slightly increasing trend in their *LAC* value in brackish water along the different cycles, even exceeding their *LAC_Max_* value in the forth–fifth cycle of operation. Moreover, the *LAC* values for both fabrics are higher than those obtained only with distilled water. All these effects may be because the precipitated salts in the fabrics act as a hygroscopic source, causing the fabric and the hydrated salts to increase their *LAC* above *LAC_Max_* slightly and above the results with distilled water considerably.

The PES sample shows the same increasing trend as the Jut and LPLA fabrics with the difference that it stabilizes its *LAC* values from the third cycle of operation around 790% absorption values. In this sample, it has been observed how the amount of salts precipitated in its structure has increased with each cycle of operation, causing the fabric to absorb a greater amount of water and salts in the time it remains in contact with the solution. Figure 6 shows the deposition of salts on the non-woven structure of the material in the red circles. These precipitated salts act as a bonding bridge with the solution, creating polar bonds through the material’s porous structure, allowing water to penetrate its structure and making the material absorb water quickly.

Compared with the exposure of the samples to distilled water, all of them have improved their absorption capacity and have come closer to their maximum value when exposed to a brackish water solution. In view of an industrial application, the most interesting fabric in the exposure to brackish water has turned out to be the PES, since from its third cycle of operation it has shown *LAC* values close to its maximum value and remains stable in subsequent work cycles.

As can be seen in the results obtained, the fabrics have improved their absorption capacity when exposed to different operation cycles in a brackish water solution. By applying these cycles of operation to the fabrics, the deposition of salts on their structure has been observed, causing an improvement in the more porous and thicker fabrics with respect to their exposure only to distilled water. The effect of these salts acted as an ionic bridge between the structure of the material and the water in the solution, allowing it to penetrate the interior of the structure during the time that each of the samples was submerged.

Finally, based on the obtained results, it can be concluded that lignocellulosic and synthetic fibers can be an industrial alternative for treating brackish water in an evaporation unit for zero liquid discharge (ZLD). As observed, the absorption capacity of brackish water by the fibers increases as the operating cycles increase. This increase in absorbent capacity could have positive industrial implications as it would allow reducing the evaporation surface of treatment units. Only the solid residue formed by salts would remain as waste. On the other hand, it is crucial for their industrial application that the mechanical properties of the fabrics withstand prolonged working cycles.

### 3.6. Mechanical Characterization of Fabrics

The results obtained from the tensile test can be seen in Table 5 for the fabrics in dry conditions and Table 6 for the fabrics in wet conditions.

As can be seen in the results obtained, the WL-WT fabric shows a deficient mechanical behavior in dry conditions; it does not show any resistance in wet conditions. This performance is due to the fact that this fabric is composed of 70% palm pruning waste, 10% lyocell and 20% PLA without applying a thermal treatment. Once calendaring is carried out, the PLA improves mechanical properties as a compatibilizer with the fibers. This effect can be seen compared with the WL-T properties shown in Table 5, which go from tensile strength values of 0.11 MPa to 0.73 MPa, and tensile modulus values from 0.010 MPa to 73.09 MPa. Faruk et al. [46] showed tensile strength values for Sisal leaf fibers of 511 MPa and for pineapple of 400 MPa. This leads to the conclusion that the WL-WT sample is unsuitable for industrial use due to its low mechanical properties. The values obtained in the present study differ from those reported in the literature mainly because they were performed on fibers and not on a fabric structure. For the elongation at break, Sanjay et al. [47] reported values like those obtained for the WL-T sample.

In dry conditions, the natural fiber fabric with the highest tensile strength was the Jut sample, with 20.45 MPa. This tensile strength value is lower than reported in the literature; however, the tensile modulus, with a value of 43.01 MPa, presents values similar to those shown by Abiola et al. [48]. The LPLA and Bam samples have lower tensile strength values than the Jut fabric, but their tensile modulus and elongation at break values are higher. These values are bigger than those reported by authors such as Chokshi et al. [18], who give elongation at break values ranging from 1.30 to 7.00% and tensile modulus ranging from 11.00 to 35.91 MPa. On the other hand, De Rosa et al. [49] obtained elongation at break values from 2.26 to 5.04%, while in the present study a value of 22.40% has been registered for LPLA. This effect may be due to the PLA present in the sample, since it is found in a proportion of 50% and depends on its orientation, as demonstrated by Hanon et al. [50].

After exposure to wet test conditions, Jut, LPLA, and Bam natural fabric samples experimented an improvement in all the mechanical properties measured in this work. Except for WL-WT, all-natural fabrics increased their elongation at break. The most remarkable cases are WL-T, which increased from 2% in dry conditions to 26.1% in wet conditions, and LPLA, which increased from 22.4% to 49.14%.

These results differ from those obtained with synthetic fabrics. In the case of the GF-NW sample, the cohesion between its fibers is very low and its mechanical properties could not be determined because its structure does not remain stable during the tensile test. On the other hand, the GF-W fabric shows higher mechanical properties than the rest of the fabrics tested in this study. Its tensile strength in dry conditions is 405.2 MPa. Mohanty et al. [51] reported values for glass fiber used in composite materials from 2000 to 3500 MPa. These values are much higher than those reported in the present study, probably because the fibers are not the same size and have not been studied in fabric form. However, the elongation at break measured by Mohanty et al. was 2.5%, while in the present study it is 20.8%, due to the low thickness and density of the sample. The results by Mohanty et al. for the Ara fabric show values of 3000 to 3150 Mpa of tensile strength and an elongation at break between 3.3 and 3.7%. These high mechanical properties are due to its high chemical stability. Finally, the PES fabric shows low mechanical properties, except for the elongation at break in dry conditions with a value of 31.33% and 182.9% in wet test conditions. The elongation at break for the PES sample and the tensile modulus for the GF-W sample are the only mechanical properties with a behavior similar to that of natural fabrics, since they increase when passing to wet test conditions. A worsening has been observed for the remaining mechanical properties studied in the synthetic materials from a test in dry to wet conditions.

Methacanon et al. [52] studied the mechanical behavior of different natural fibers, including sisal, roselle or reed. Their studies showed that when fibers were subjected to tensile stress in dry and wet conditions the tensile strength was not seriously affected, with values between 150 and 200 MPa for the sisal sample. However, there is a noticeable difference in the elongation at break in both test conditions, with the difference between all samples being at least 5%. Their paper shows the same behavior as that obtained in the present study, where the samples tested in wet test conditions show an improvement in their mechanical properties and elongation at break. Nevertheless, Athijayamani et al. [53] conducted a similar study with sisal fibers focused on the mechanical properties of a composite material with different fiber content and lengths. The results showed a decreasing trend in tensile strength when the materials were exposed to wet conditions. Finally, Ramamoorthy et al. [45] studied composite materials using jute fibers. The results showed the same tendency as other authors using natural fibers as a reinforcement material in composite materials in dry test conditions, obtaining better mechanical properties than in wet conditions. This effect is due to the fact that absorbing water creates a separation between the fabrics and the polymeric matrix, which causes fragility in the composite material. When studying the natural fibers without the polymeric matrix, mechanical properties have been improved under wet conditions, as demonstrated by Methacanon et al., in agreement with the results obtained in the present investigation.

In view of the results obtained in tensile test between natural and synthetic fabrics, it can be observed that for, the practical application that requires the use of these fabrics for the treatment of brackish water, synthetic fabrics stand out above natural fabrics. In an industrial application, such as a vertical arrangement of fabrics for an evaporation unit, synthetic fabrics would be more interesting as they could withstand longer working cycles.

### 3.7. Thermal Characterization of Textiles

The results of the thermogravimetric test can be seen in Figure 7. Most samples lose a significant mass between 300 and 400 °C, except for the Ara sample, where significant mass loss occurs between 550 and 650 °C. On the other hand, the GF-W sample has not been included in Figure 7a as its behavior is similar to that of the GF-NW sample and does not provide relevant information. This is because the sample loses less than 5% of its mass in the selected temperature range. Samal et al. [54] demonstrated a similar behavior obtained for glass fibers with a small mass loss at 350 °C, but with a total loss of less than 5%. This performance is due to the thermal stability of glass fiber below 1000 °C, as Laoubi et al. [55] mentioned in their research on glass fiber composite material with polyester resins.

In Figure 7a, the lignocellulosic fabrics have the same behavior; the loss of mass of the PES fabric can be highlighted because its change occurs in the same temperature range of 300 to 400 °C, but it does not lose its entire mass and stabilizes at the end of the test at around 35% together with the Ara sample. These two samples’ degradation is incomplete, as they reach a residual value of approximately 35%. This may be because the fabric samples used have some kind of inorganic charge in their structure that does not degrade in the working temperature range.

Figure 7b shows three clearly differentiated zones in the behavior of the fabrics for the chosen temperature ramp. A slight variation in the samples is observed in the first section, where the temperature changes from 50 to 180 °C. This change is due to the evaporation of the ambient water contained in the samples. In the range from 180 to 200 °C they no longer undergo any change, indicating that no water remains. In Section 3.1 Humidity in fabrics, the humidity contained in the samples was determined using a precision balance and an oven to evaporate the water. Comparing both results, it can be observed that the GF-NW and PES fabrics do not produce any change, as shown in the enlarged image in Figure 7b, and the remaining fabrics show similar behavior to that obtained in Section 3.1 where the WL-WT, WL-T, and Jut samples are the ones with the highest humidity content. In the 200 to 500 °C range, a loss of mass of the fabrics is observed, except for the GF and Ara samples. As can be seen, the maximum rate of degradation of the fabrics ranges between 350 and 400 °C. On the other hand, the last differentiated zone corresponds to the 500 to 700 °C range where the Ara sample degrades. Due to its stability, it presents a slow degradation rate in the 400 to 500 °C range, becoming more severe from this temperature onwards and presenting its maximum degradation rate at around 600 °C.

Table 7 shows the main results obtained from the thermogravimetric analysis of natural and synthetic fabrics. In this table, the three sections mentioned above have been analyzed. In the first part, where the *T_onset_* is set at 50 °C and the *T_endset_* at 180 °C, the loss of mass corresponding to the evaporation of water retained in the fabrics by the environmental humidity is observed. Comparing these results with those obtained in Section 3.1, the values obtained in both experimental trials are similar, since the WL-WT and WL-T fabrics obtains values of 4.28% and 5.17%, respectively, compared to the values of 3.67% and 3.87% indicated in Table 7. The values of the LPLA, Bam, and Ara fabrics are also similar to those obtained in the humidity determination. The most outstanding case is the Jut fabric which presents a difference between both tests, with 8.58% in the humidity determination compared to 4.95% in the thermogravimetric analysis. But, the difference is not very significant. Finally, it can be seen that GF-NW, GF-W, and PES fabrics have values below 1%. So, it can be concluded that this initial mass loss in the TGA analysis corresponds to the humidity in the fabrics.

The temperature value at which the maximum mass loss rate of the tested samples occurs in the second degradation step can be observed. In the case of WL-WT and WL-T fabrics, the value is very close at 362.13 and 361.65 °C, respectively. Their mass loss in this second analysis section is 84.69% for the WL-WT sample and 83.72% for the WL-T sample. Poletto et al. [56] collected thermal decomposition data of different natural fibers. Sisal, a fiber similar to palm debris from the WL samples, was observed to have a maximum degradation temperature of 347 °C. This difference of approximately 15 °C may be due to the presence of PLA in the samples, which would increase its degradation temperature. Dominguez-Candela et al. [57] worked with epoxidized chia seed oil in a PLA polymeric matrix, showing its maximum degradation temperature at 390.4 °C.

This behavior described for the WL-WT and WL-T samples also applies to the LPLA fabric, where its maximum degradation temperature occurs at 368.89 °C. The results demonstrated by Barneto et al. [58] show a degradation peak of flax fibers in the nitrogen atmosphere in the 326 to 363 °C band. Van De Velde and Kiekens [59] quantified the maximum degradation peaks ranging from 307.29 to 348.99 °C for different flax fibers with several treatments. In this case study, there was a slight increase in the degradation temperature due to the fabric’s 50% flax and 50% PLA composition.

The Jut and Bam samples are of natural origin and show maximum degradation peaks at 374.62 and 378.40 °C, respectively. Zhang et al. [60] reported a maximum degradation peak at 368 °C for untreated bamboo fibers, and 374 °C for treated fibers. On the other hand, the Jut sample has a slightly higher temperature value than that demonstrated by other authors [56] with a peak degradation value at 365 °C.

Related to the synthetic fabrics tested, we can highlight that the PES and Ara fabrics do not degrade completely, as both maintain a residue of approximately 35%. When analyzing the maximum degradation peak for PES, this occurs at 373.77 °C. Rimbu et al. [61] studied fabrics from the textile industry using zinc oxide solutions. In that study, their polyester fabrics were stable up to 380 °C. On the other hand, the Ara fabric has a maximum decomposition at a temperature of 589.96 °C because its chemical structure is more stable than the rest of the materials. Liu and Yu [62] carried out thermal decomposition investigations on aramid fabrics from lower to higher strength, obtaining temperature degradation values from 527.8 to 541.0 °C. These values are lower than those obtained experimentally in this work and are possible due to the inorganic loading that may be present in the tested samples.

### 3.8. Capillarity of Fabrics

The results obtained from the capillarity test can be seen in Figure 8 for the different fabrics used. The fabric with the fastest time to reach 135 mm of maximum height is the Ara fabric, in less than 50 min. This effect is due to the hydrophilic character of the fabric, as demonstrated in Section 3.2 Wetting contact angle measurements, where the drop of distilled water on contact with the fabric was immediately absorbed. Tran et al. [63] determined a contact angle for recycled Kevlar fibers of 23.1°, thus indicating their hydrophilic character. In addition to this effect, the rapid rise in water through the fabric may be due to the volume of hollow fibers in the sample.

The Jut fabric is the second sample to reach the maximum height in 2 h. Pinto et al. [64] investigated composite materials using jute fibers with different surface treatments. They performed a capillarity analysis on jute fibers, reporting that the untreated fibers reached a height in distilled water of 90 mm in approximately 2 h. As can be seen, the results are similar to those obtained in the present study despite using a complete jute fabric.

The third fabric with a high capillarity behavior, as seen in Figure 8, is the GF-W sample which reaches the maximum height in 24 h. The GF-NW sample has a similar behavior, but the low cohesion between its fibers means that the water cannot rise more than 105 mm in a time of 48 h. These samples have previously shown not to be able to absorb large amounts of water into their structure, but their affinity for water is very high. Testoni et al. [65] studied the capillary behavior of flax fibers and glass in n-hexane and water, determining that glass could retain more water by capillarity. Considering the results obtained in this research, they agree with those reported by Testoni et al., since the LPLA sample has risen a total of 112 mm in 48 h. Due to its water affinity and fiber volume, it is between the GF-W and GF-NW samples.

The Bam sample reaches a height of 80 mm within 40 min and remains at this height even after 48 h. Despite being a very hydrophilic fabric and having a good water absorption capacity, it has a significantly reduced thickness, so a balance is established between the height reached and the water evaporation rate, making it impossible for the water to achieve a greater height in the sample.

PES fabric is one of the slowest ones for the capillarity test. This is due to its low affinity for water, as it is a hydrophobic fabric. Ferrero [66] researched polyester and acrylic fabrics with plasma surface treatment in air, nitrogen, and oxygen, reporting that the polyester fabric without plasma treatment reached a height of less than 50 mm. This behavior is similar to that demonstrated in the present study and the height difference may be due to the thickness of the sample used and the cleaning treatment they applied.

Finally, the WL-T and WL-WT samples have shown the worst performance in the capillarity test. The WL-T sample reaches a maximum height of 43 mm and the WL-WT sample reaches 7 mm. The difference between the two fabrics is mainly due to the thickness of the samples. As the one with the heat treatment has a lower thickness, the weight of water rising through the fabric is lower. On the other hand, reducing the number of free fiber voids makes it easier for water to interact between fibers. Yanılmaz and Kalaoğlu [67] investigated the behavior of acrylic fabrics, observing that the thickness of the samples influenced their capillarity performance, demonstrating that a lower thickness can improve the capillarity of the materials.

The effect of fabrics on capillarity is very diverse and depends on several factors. Ara and Jut fabrics would be the most favorable for an industrial application, as they show the highest capillarity in the shortest time in contact with water. This effect favors that the whole fabric is wet in contact with the problem solution; therefore, there is greater availability for subsequent evaporation. On the other hand, the WL-WT fabric would be discarded, as it hardly shows sufficient capillary effect.

### 3.9. Samples under Composting Conditions

The results obtained in the degradation experiment under composting conditions can be seen in Figure 9. The LPLA sample loses 62.22% of its mass in the first sampling at 14 d. This severe loss of mass in the first extraction of the sample is due to PLA degradation, as the sample is composed of 50% PLA. The mass loss is also visible in the first image presented in Figure 10, showing that the fabric has lost considerable thickness and there are gaps between its fibers, where the initial material had no visible gaps. The second sample is taken after 30 d, where the fabric has lost 76.82% of its initial weight, corresponding to the total degradation of the PLA and part of the lignocellulosic flax fibers. These lignocellulosic fibers are more stable and almost maintain their behavior throughout the extraction of the remaining samples. The last extraction at 112 d has lost 86.34% of its initial mass. Figure 10 shows how the LPLA sample has deteriorated to fragmentation with a slight manipulation. Dominguez-Candela et al. [57] investigated the effects of epoxidized chia seed oil on PLA composite materials. Their research showed that virgin PLA was completely degraded at approximately 27 d. This effect on PLA would account for the degradation of the LPLA sample described in this research. On the other hand, Lerma-Canto et al. [68] studied the effect on thermoplastic materials with PLA. In their research one of their composite materials contained 80% of PLA, observing that it was wholly degraded in 24 d.

This behavior is similar for WL-WT and WL-T samples, where in the first 30 d they lost 46.78 and 49.62%, respectively. This can be attributed to the cleaning and sample composition of 20% PLA and 10% Lyocell. Contrary to the LPLA sample in Figure 10, no visible degradation is observed during the extraction of the samples. These samples lost most of their mass due to the PLA in combination with Lyocell, in addition to the loss due to the fabric’s fibers during cleaning because of their poor mechanical cohesion. A stabilization is observed in the extraction of the fourth sample, reaching a degradation of 68.16% in the WL-WT sample and 70.75% in the WL-T sample for a time of 112 d.

The Jut fabric shows a slower degradation behavior initially, where in the first extraction of the sample at 14 d it degraded 9.11%. From the second extraction onwards, its degradation is more affected, degrading 79.67% of its initial weight at 112 d; this is more than the WL-WT and WL-T samples. Lau et al. [69] investigated the degradation of EcoPLA composites with cotton and jute fibers. Their results showed that jute ropes degraded for 105 d by 84.8%. This is a similar result to the one obtained in this research despite being a different sample and composting configuration. Figure 10 shows the deterioration in the Jut sample, where the fibers in the fabric shrink with each extraction and its structure has broken down by the last sample.

Finally, the Bam sample is the fabric that has shown the worst behavior, since for a time of 112 d in the extraction of the last sample it has degraded by 5.12%. This degradation may be due to the cleanliness of the samples, as no visible deterioration can be seen in the extracted samples in Figure 10. Satyanarayana et al. [70] showed that the environmental degradation in composite materials varies depending on the type of natural fiber used. Their paper reported that a bamboo pole has a degradation time in the environment of between 1 to 3 years. This environmental degradation time shows how resistant the material is to environmental conditions, which is consistent with the fact that the Bam fabric did not degrade in this research.

## 4. Conclusions

From the results obtained in the present investigation for the characterization of lignocellulosic and synthetic fabrics, it has been observed that the WL-WT, GF-NW, and GF-W samples would be discarded for industrial use in the treatment of complex effluent. The *LAC* shown in distilled water and the *LAC* per duty cycle is insufficient in the GF-NW and GF-W samples. On the other hand, WL-WT showed an excellent *LAC* in distilled and brackish water, but its low mechanical resistance made its use at the industrial level not advisable.

The PES and WL-T fabrics showed the best behavior for industrial use in brackish water treatment, as their *LAC* stabilized after the 3rd operating cycle due to the precipitation of salts in their structure, which facilitated water absorption, showing *LAC* values of 316.35 and 789.02%, respectively.

In addition to these fabrics, the behavior of the Bam and Ara samples stands out. Bam has shown acceptable *LAC* and mechanical characteristics for industrial use, as its thickness is small compared to the rest of the fabrics. On the other hand, LPLA and Jut fabrics show similar *LAC* behavior to Bam fabric with brackish water but show slightly higher salt retention in their structure. The high production of these natural fabrics shows that they can be interesting for industrial use. Finally, the Ara fabric has shown excellent hydrophilic behavior, with a *LAC* value over time similar to the *LAC_Max_* value. In addition to this, in view of an industrial application where the fabrics must be fixed without movement, it has shown the best capillary behavior for water treatment.

## Figures and Tables

**Figure 1 polymers-16-00084-f001:**
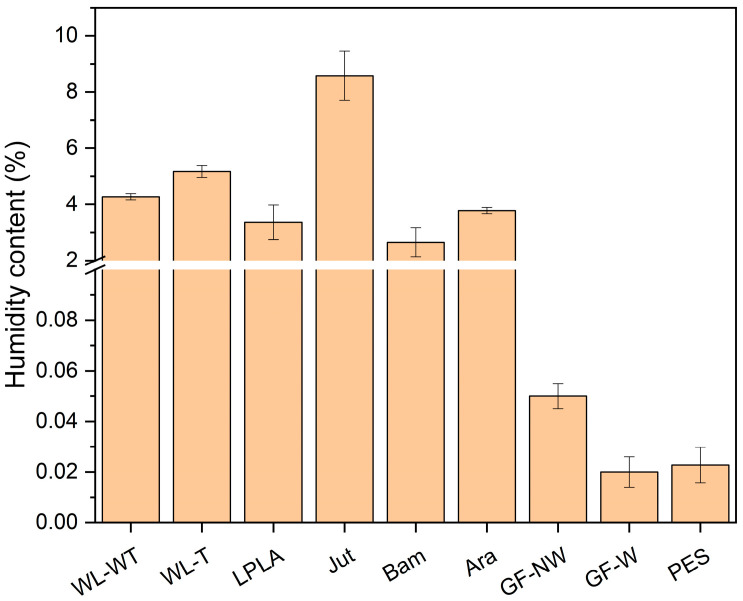
Humidity content of fabrics.

**Figure 2 polymers-16-00084-f002:**
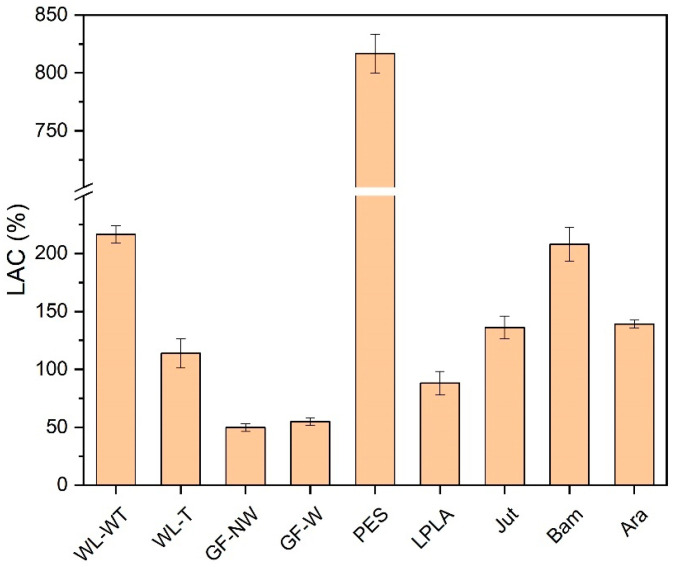
Distilled water absorption capacity in test fabrics.

**Figure 3 polymers-16-00084-f003:**
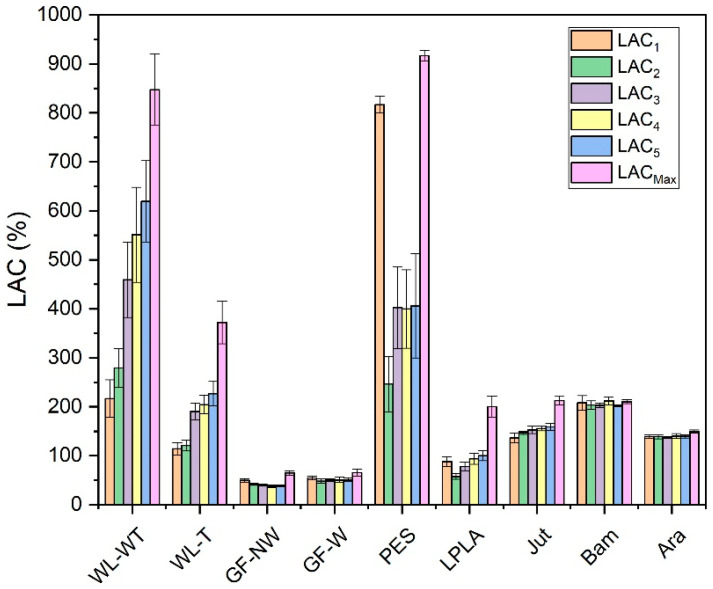
*LAC* obtained for five absorption and drying cycles with distilled water.

**Figure 4 polymers-16-00084-f004:**
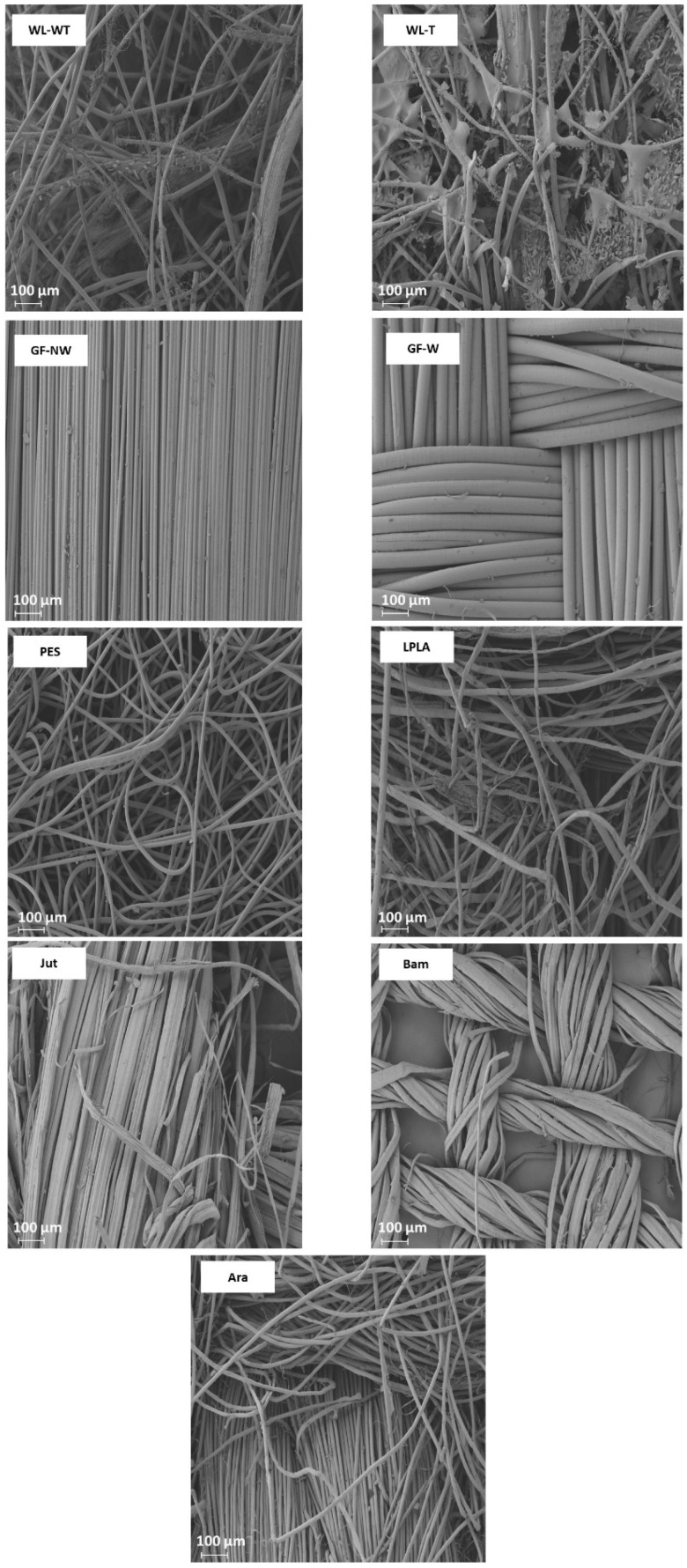
FESEM images of the fabrics used in the determination of *LAC* in distilled water.

**Figure 5 polymers-16-00084-f005:**
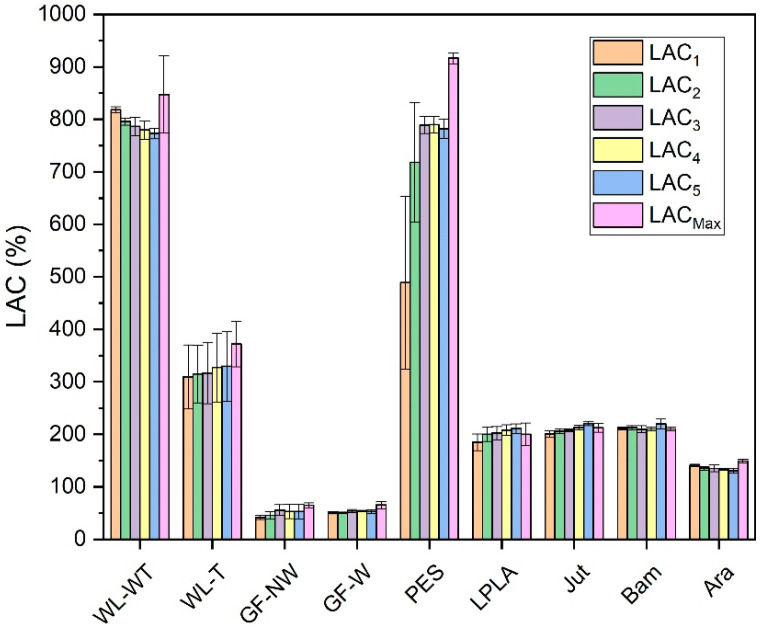
*LAC* for five absorption and drying cycles in brackish waters.

**Figure 6 polymers-16-00084-f006:**
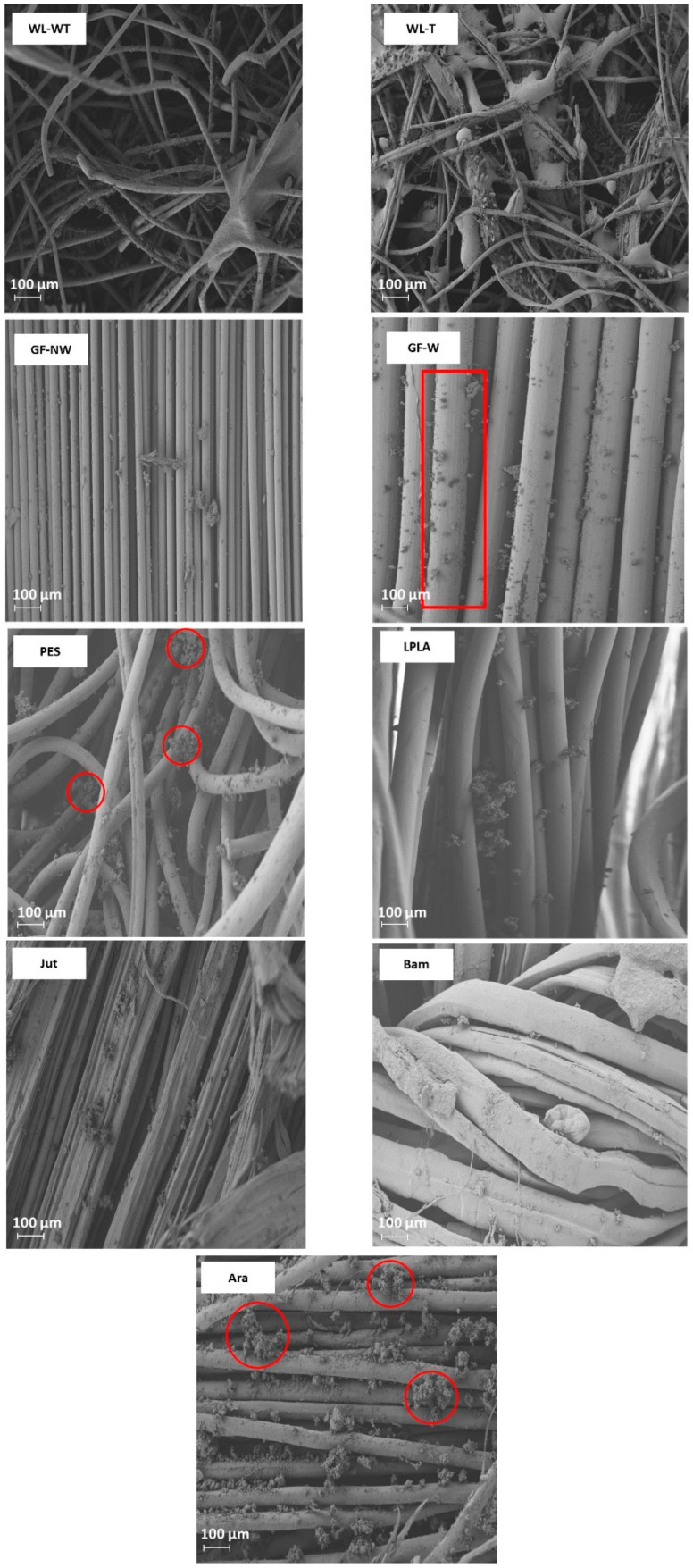
FESEM images of the surface of the fabrics once they have been exposed to the brackish water solution after the five work cycles.

**Figure 7 polymers-16-00084-f007:**
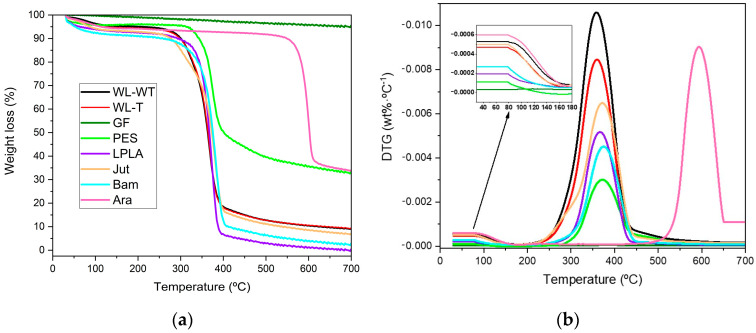
Thermal parameters of degradation of natural and synthetic fabrics: (**a**) Weight loss; (**b**) Differential thermogravimetry.

**Figure 8 polymers-16-00084-f008:**
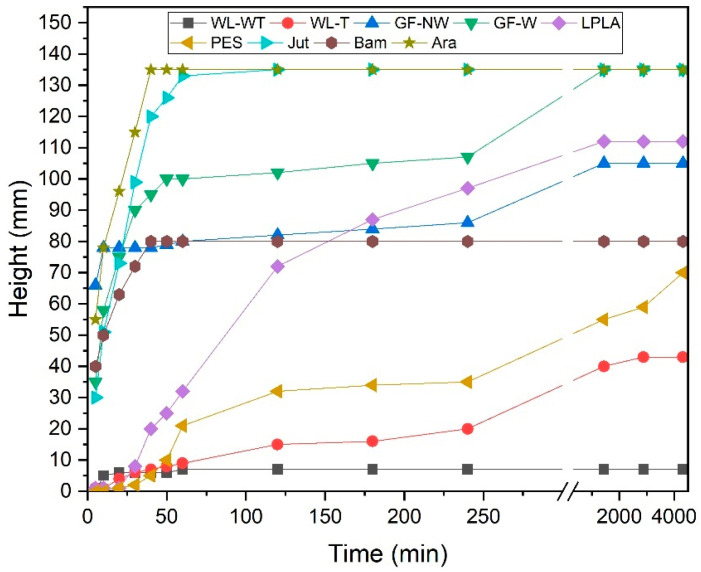
Capillarity of samples in distilled water for a maximum height of 135 mm.

**Figure 9 polymers-16-00084-f009:**
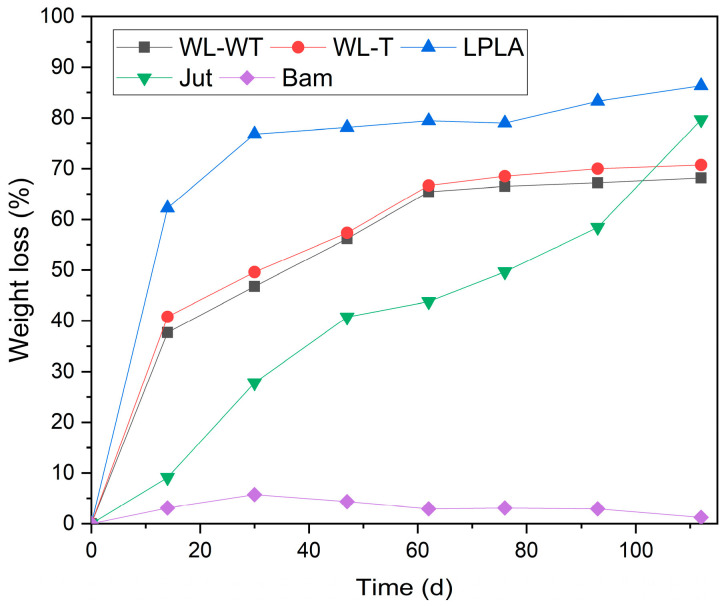
Degree of disintegration via composting for samples over a period of 112 days.

**Figure 10 polymers-16-00084-f010:**
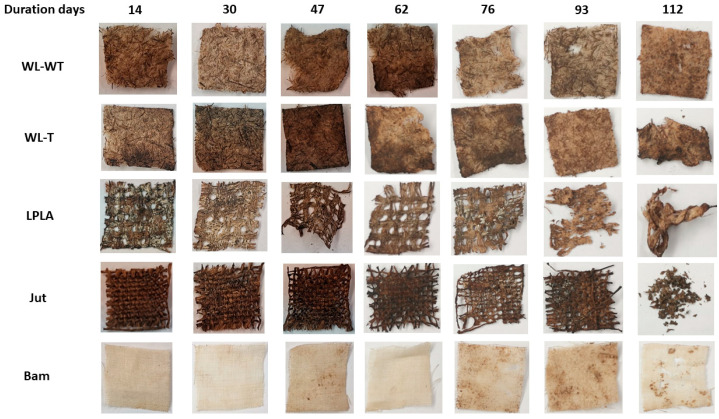
Visual appearance of samples under composting conditions.

**Table 1 polymers-16-00084-t001:** Main characteristics of the natural fabrics.

Natural Fabrics	Cellulose (%)	Hemicellulose (%)	Lignin (%)	Waxes (%)	References	Fabric Structure	Thickness (mm)	Grammage (g·m^−2^)
WL-WT	43–78	10–24	3.75–11	2	[25,26]	Non-woven	2.0	273
WL-T						Non-woven	0.5	330
LPLA	71–81	16.7–21	2–3	1.5–1.7	[25,27,28]	Taffeta	0.6	556
Jut	45–72	13–21	12–26	0.5	[26,27,29]	Taffeta	0.7	354
Bam	26–74	12–74	21–31	0.9	[25,28]	Taffeta	0.2	56

**Table 2 polymers-16-00084-t002:** Main characteristics of the synthetic fabrics.

Synthetic Fabrics	Fabric Structure	Thickness (mm)	Grammage (g·m^−2^)
GF-NW	Unidirectional	0.5	600
GF-W	Taffeta	0.3	196
PES	Non-woven	0.8	212
Ara	Taffeta	1.0	572

**Table 3 polymers-16-00084-t003:** Concentrations of salts used to synthesize brackish water.

Salts	Concentration (mg·L^−1^)
NaNO_3_	538.018
KNO_3_	28.587
Na_2_B_4_O_7_·10H_2_O	1.265
CaCl_2_	573.264
Ca(OH)_2_	544.797
MgSO_4_·7H_2_O	3144.773
NaHCO_3_	1390.668
CuSO_4_·5H_2_O	0.060
LiCl	1.620

**Table 4 polymers-16-00084-t004:** Wetting angle measurements on different fabrics.

Sample	*Θ* _0_	*Θ_f_*	*t* (min)	∆*Θ*·(min^−1^)
WL-WT	114.6	0	9	12.73
WL-T	122.1	0	14	8.72
GF-NW	-	-	-	-
GF-W	-	-	-	-
LPLA	152.7	128.6	35	0.69
PES	129.9	96.5	20	1.67
Ara	-	-	-	-
Jut	156.5	0	4	39.13
Bam	-	-	-	-

**Table 5 polymers-16-00084-t005:** Mechanical properties of fabrics in dry conditions.

Fabric Sample	Tensile Strength (MPa)	Tensile Modulus (MPa)	Elongation at Break (%)
WL-WT	0.11 ± 0.02	0.010 ± 0.001	2 ± 0.01
WL-T	0.73 ± 0.05	73.09 ± 0.23	2 ± 0.02
GF-NW	-	-	-
GF-W	405.2 ± 133.3	151.2 ± 24.1	20.8 ± 1.1
PES	8.05 ± 0.4	10.4 ± 0.8	31.33 ± 2.3
LPLA	6.34 ± 1.1	47.43 ± 7.9	22.4 ± 7.9
Jut	20.45 ± 2.7	43.01 ± 4.4	20.5 ± 2.5
Bam	3.18 ± 1.7	64.85 ± 36.3	44 ± 5.5
Ara	91.95 ± 14.2	289.5 ± 48.0	29.6 ± 2.2

**Table 6 polymers-16-00084-t006:** Mechanical properties of fabrics in wet conditions.

Fabric Sample	Tensile Strength (MPa)	Tensile Modulus (MPa)	Elongation at Break (%)
WL-WT	-	-	-
WL-T	0.49 ± 0.02	30.95 ± 0.36	26.1 ± 1.3
GF-NW	-	-	-
GF-W	59.66 ± 20.2	726.7 ± 174.9	19.18 ± 4.8
PES	2.14 ± 0.2	0.59 ± 0.1	182.9 ± 16.2
LPLA	13.93 ± 1.3	67.39 ± 15.8	49.14 ± 10.0
Jut	23.63 ± 1.8	117.5 ± 26.9	22.71 ± 7.5
Bam	16.32 ± 6.1	49.8 ± 33.5	51.95 ± 20.5
Ara	19.85 ± 5.29	108.6 ± 22.56	19.28 ± 4.17

**Table 7 polymers-16-00084-t007:** Summary results of TGA parameters of natural and synthetic fabrics.

Samples	First Step of Degradation			Second Step of Degradation			
	*T_onset_ *(°C)	*T_endset_ *(°C)	∆*wt* (%)	*T_onset_ *(°C)	*T_endset_ *(°C)	*T_max_* (°C)	∆*wt* (%)
WL-WT	50	180	3.67	200	600	362.13	84.69
WL-T	50	180	3.87	200	600	361.65	83.72
GF-NW	50	180	0.77	200	600	-	3.01
GF-W	50	180	0.37	200	600	-	3.21
PES	50	180	0.84	200	600	373.77	60.57
LPLA	50	180	3.19	200	600	368.89	91.16
Jut	50	180	4.95	200	600	374.62	84.22
Bam	50	180	4.22	200	600	378.40	87.30
Ara	50	180	3.93	400	700	589.96	58.69

## Data Availability

The data presented in this study are available upon request from the corresponding author.

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
