# Peer review of "Characterization of Natural and Synthetic Fabrics for the Treatment of Complex Wastes"

_polymers, 2023, doi:10.3390/polym16010084_

Round 1

Reviewer 1 Report

Comments and Suggestions for Authors

The introduction is clear in presenting the study's objective, which is testing 9 fabrics for brackish water treatment with an industrial application focus under the concept of zero liquid discharge (ZLD). However, here in the following some comments

- The mention of moisture content in lignocellulosic fabrics and the observation of hydrophobicity in LPLA are informative. It would be helpful to briefly explain the significance of these observations or how they relate to the overall goal of brackish water treatment.

- The comparison of liquid absorption capacity between synthetic and natural fabrics is a key aspect. It might be beneficial to provide insights into the practical implications of these findings, especially in the context of brackish water treatment and zero liquid discharge.

- The note about the synthetic fabrics being more resistant but still competing in terms of applicability is intriguing. Consider providing more details on how these characteristics might influence their use in industrial applications.

-An abbreviation list should be provided.

Comments on the Quality of English Language

Minor editing of English language are required.

Reviewer 2 Report

Comments and Suggestions for Authors

Dear Authors

With the results obtained in the present investigation for the characterization of lignocellulosic and synthetic fabrics, it has been observed that the WL-WT, GF-NW, and GF-W samples would be discarded for industrial use in the treatment of complex effluent. The LAC shown in distilled water and the LAC per duty cycle are insufficient in the GF-NW and GF-W samples. On the other hand, WL-WT showed an excellent LAC in distilled and brackish water, but its low mechanical resistance made its use at the industrial level not advisable. The PES and WL-T fabrics showed the best behavior for industrial use in brackish water treatment, as their LAC stabilized after the 3rd operating cycle due to the precipitation of salts in their structure, which facilitated water absorption, showing LAC values of 316.35 and 789.02% respectively. In addition to these fabrics, the behavior of the Bam and Ara samples stands out. Bam has shown acceptable LAC and mechanical characteristics for industrial use, as its thickness is small compared to the rest of the fabrics. On the other hand, LPLA and Jut fabrics show similar LAC behavior to Bam fabric with brackish water but show slightly higher salt retention in their structure. The high production of these natural fabrics shows that they can be interesting for industrial use. Finally, the Ara fabric has shown excellent hydrophilic behavior, with an LAC value over time similar to the LACMax value. In addition to this, given an industrial application where the fabrics must be fixed without movement, it has shown the best capillary behavior for water treatment.

General comments

It is hard to find out the novelty of the presented work. The work is closer to being a "Review" than a "Research" article. All the presented results are just measures of the physicochemical properties of untreated fabrics. No modification has been applied to the fabrics to enhance their physicochemical properties correlated to brackish water treatment. No measures of the removed contaminates from the brackish water have been presented. Even, no analysis of the brackish water composition before and after being treated with the selected fabrics has been provided.

Specific comments 

In the abstract section, the authors mentioned "LPLA, PES, WL-WT, and PLA" without a full description. Please mention the full name for any abbreviation when appears for the first time in the text.  

Comments on the Quality of English Language

A minor revision is required. 

Round 2

Reviewer 2 Report

Comments and Suggestions for Authors

Dear Authors

Thanks for the clarifications. 

Greetings

Comments on the Quality of English Language

A minor revision of the language is ecommended.